# A One Health Approach to Combatting *Sporothrix brasiliensis*: Narrative Review of an Emerging Zoonotic Fungal Pathogen in South America

**DOI:** 10.3390/jof6040247

**Published:** 2020-10-26

**Authors:** John A. Rossow, Flavio Queiroz-Telles, Diego H. Caceres, Karlyn D. Beer, Brendan R. Jackson, Jose Guillermo Pereira, Isabella Dib Ferreira Gremião, Sandro Antonio Pereira

**Affiliations:** 1Mycotic Diseases Branch, Centers for Disease Control and Prevention, Atlanta, GA 30329, USA; mwi4@cdc.gov (J.A.R.); xju7@cdc.gov (D.H.C.); ydh7@cdc.gov (K.D.B.); 2Epidemic Intelligence Service, Centers for Disease Control and Prevention, Atlanta, GA 30329, USA; 3Department of Public Health, Hospital de Cíinicas, Federal University of Parana, Curitiba 82015-154, Brazil; queiroz.telles@uol.com.br; 4Center of Expertise in Mycology, Radboudumc/CWZ, 6532 SZ Nijmegen, The Netherlands; 5Ministry of Public Health and Social Welfare, National Leprosy Control Program, National Directorate of Health Surveillance, Dermatology Specialty Center, San Lorenzo 2160, Paraguay; jose_pereira15@hotmail.com; 6Laboratory of Clinical Research on Dermatozoonoses in Domestic Animals, Evandro Chagas National Institute of Infectious Diseases, Oswaldo Cruz Foundation (Fiocruz), Rio de Janeiro 21040-360, Brazil; isabella.dib@ini.fiocruz.br (I.D.F.G.); sandro.pereira@ini.fiocruz.br (S.A.P.)

**Keywords:** sporotrichosis, *Sporothrix brasiliensis*, cat-transmitted sporotrichosis, One Health, fungi

## Abstract

Cat-transmitted sporotrichosis caused by *Sporothrix brasiliensis* has become a major public health concern and presents a distinct divergence from the traditional epidemiology of sporotrichosis. This emerging fungal pathogen spreads readily among cat populations, and human infections occur exclusively via zoonotic transmission. While sporotrichosis is an implantation mycosis that typically manifests as cutaneous lesions in humans and cats, severe extracutaneous manifestations are more common with *S. brasiliensis* than other *Sporothrix* species infections. Rapid diagnosis and appropriate treatment regimens are critical for successful clinical resolution of sporotrichosis in both cats and humans. Species-level identification of *Sporothrix* is possible with molecular diagnostics and necessary for tracking the geographic expansion of *S. brasiliensis* and better understanding its epidemiology. Combatting cat-transmitted sporotrichosis requires a One Health approach to successfully implement public health control measures.

## 1. Introduction

Sporotrichosis caused by *Sporothrix brasiliensis* is not the ordinary “Rose Gardener’s Disease” typically encountered in North America and Europe. *Sporothrix brasiliensis* is instead an emerging fungal pathogen with cat-to-human (zoonotic) and cat-to-cat/dog transmission (enzootic) and epidemic and epizootic potential [1].

Fungi in the genus *Sporothrix* are typically found in the environment, associated with soil, plants, and decaying wood. Most *Sporothrix* spp. do not cause human or animal infections. However, the “pathogenic clade” of the genus *Sporothrix*, which includes *S. schenckii*, *S. globosa*, and *S. brasiliensis*, infects mammals [1,2,3]. Sapronotic transmission (from the environment) was the most common source of human sporotrichosis historically, but zoonotic infections have become increasingly common with the emergence of *S. brasiliensis*. Sporotrichosis can present with cutaneous and extracutaneous manifestations, and infection severity differs depending on infecting species and host immune status [4,5,6,7,8]. Whereas transmission of *S. schenckii* typically occurs through traumatic inoculation of the fungus via plant matter, or less frequently, by inhalation of the mold form, *S. brasiliensis* is almost exclusively transmitted via the bite, scratch, or contact with the exudate of cutaneous lesions of an infected cat (*Felis catus*) [9,10,11].

*Sporothrix brasiliensis* has gone from a localized clinical oddity two decades ago to a major public health concern, especially in Brazil. It has caused tens of thousands of cases across wide geographic areas and has spread through cat-to-cat and zoonotic transmission. The first epidemic of cat-transmitted sporotrichosis (CTS) caused by *S. brasiliensis* was detected in 1998, in Rio de Janeiro, Brazil. At that time, feline sporotrichosis had been reported in 3 states in the South and Southeast regions of Brazil [9]. Currently, cases of CTS have been detected from Rio Grande do Sul in South Brazil to Rio Grande do Norte in Northeast Brazil, an area spanning over 2500 miles, and cases have begun to spread to other American countries, including Argentina, Paraguay, and Panama [12,13,14,15,16]. This emerging species of *Sporothrix* has produced a dramatic increase in the incidence of zoonotic sporotrichosis and the largest outbreak of animal sporotrichosis ever reported [17,18]. This review will examine human and feline sporotrichosis caused by *S. brasiliensis*. The first section will address the disease in humans, with general information on the epidemiology, clinical characteristics, and treatment of CTS. The next section will cover the clinical manifestations and treatment of feline sporotrichosis. We will then address techniques for diagnosing human and animal sporotrichosis, and the prevention and control of *S. brasiliensis*. Throughout this review, we will focus on differentiating *S. brasiliensis* from other pathogenic *Sporothrix* spp. and the need for a One Health approach to control the spread of cat-transmitted sporotrichosis.

## 2. Materials and Methods

We conducted a narrative literature review to improve the understanding of *S. brasiliensis* epidemiology, clinical presentation, treatment for infections in humans and cats, and the public health implications of this emerging pathogen. A systematic review was not conducted given the need for a broad review to address a range of questions. We searched 16 databases (including MEDLINE and EMBASE) for studies in English, Spanish, or Portuguese, and published since 2008 that included information about *S. brasiliensis*. The references from primary studies and reviews were also reviewed for additional relevant publications. A detailed search strategy can be found in Appendix A.

## 3. Causal Agent

Like other *Sporothrix* spp., *S. brasiliensis* is a thermal dimorphic fungus able to undergo a morphological transition from filamentous hyphae (mold form) in the environment (25 °C) to parasitic yeast form in host tissue (35–37 °C). This thermal dimorphism is thought to explain *Sporothrix* species pathogenicity in mammalian hosts [8]. *Sporothrix brasiliensis* was only recently recognized as a unique species. In 2007, scientists discovered that *S. schenckii* (discovered in 1898) was composed of multiple genetically distinct species, and proposed new clinically important *Sporothrix* species names, including *S. brasiliensis* [19,20]. One of the major differences between *S. schenckii* and *S. brasiliensis* is the increased virulence observed in the latter [21]. Examination of the cell wall shows that while both species have bilayered cell walls, that of *S. brasiliensis* has less glycoprotein 70 kDa (gp70) antigen, a thicker cell wall with greater rhamnose and chitin content, and longer cell wall microfibrils that can connect yeast cells to form biofilms. These unique characteristics contribute to the increased drug resistance and virulence [21,22].

## 4. Geographic Distribution

Human and feline infections with *S. brasiliensis* have been documented in 3 countries (Brazil, Argentina, and Paraguay), and cases in Paraguay have been linked to travel from Brazil. Although zoonotic sporotrichosis caused by other *Sporothrix* spp. has been documented more widely (United States, Malaysia, India, and Mexico), these infections were caused by *S. schenckii* and typically occur as isolated cases or small and contained outbreaks, and so will not be discussed in this paper [13,23,24,25,26,27,28,29]. However, it is important to note that some fungal identification systems may misidentify *S. brasiliensis* as *S. schenckii*, therefore, reported identifications should be interpreted with caution. The term CTS will be used to refer to cases caused by *S. brasiliensis* exclusively throughout the rest of this manuscript. Most of what is known about the geographic distribution of *S. brasiliensis* comes from Brazil. During 1998–2004, more than 750 human cases and 1500 feline cases of sporotrichosis were diagnosed by the Evandro Chagas National Institute of Infectious Diseases/Fiocruz, Rio de Janeiro, Brazil, although retrospective identification of *Sporothrix* isolates detected cases of *S. brasiliensis* in Brazil as early as 1989 [30]. Most of these cases had reported contact with an infected cat, but speciation to differentiate *S. brasiliensis* infections was not available at that time [31]. By 2011, >4100 human cases and >3800 feline cases had been diagnosed at this research institute, making this the largest CTS outbreak ever reported [18]. By 2014, >200 canine cases had also been identified [32,33,34]. The disease is considered enzootic in some animal populations, affecting hundreds of dogs (*Canus lupus familiaris*) and thousands of cats in the state of Rio de Janeiro and across South and Southeast Brazil [1,34,35]. Brazilian health authorities have now received case notifications of CTS from 11 of 25 states in four of Brazil’s five geographical regions, with unofficially reported evidence of cases in 5 additional states [1,15,35,36]. Cases of CTS have also been documented in the Buenos Aires Provinces of Argentina, with evidence of increasing cases in Buenos Aires and a suspected CTS outbreak in El Calafate, Santa Cruz Province [12,14,37]. The earliest detection of *S. brasiliensis* in Argentina came from retrospective sampling of human and animal clinical isolates from Misiones and Buenos Aires provinces in 1986 and 1988 and soil samples from Chaco Province in 2003 [12]. The known endemic region of *S. brasiliensis* is currently limited to Brazil and Argentina. The detection of *S. brasiliensis* in humans, animals, and the environment demonstrates the shift from sapronotic to zoonotic transmission and emphasizes the importance of a One Health approach to combatting this pathogen.

The travel-associated CTS cases in Paraguay occurred in 2017 among family members who moved from Brazil with an infected cat [13], and laboratory testing performed after the published report found the etiologic agent to be *S. brasiliensis* (unpublished data from Paraguay Ministry of Public Health and Social Welfare). This is the first record of a travel-associated case of *S. brasiliensis* outside of Brazil or Argentina, documenting the potential for spread to new areas through transport of cats. A case report from the United States provides additional support for this potential route of spread, detailing how a Brazilian woman was diagnosed with CTS caused by *S. schenckii* at a hospital in Boston, Massachusetts, following contact with an ill cat in Minas Gerais, Brazil [38].

## 5. Animal Hosts, Reservoirs, and Sources of Human *S. brasiliensis* Infection

Human-to-human transmission of *S. brasiliensis* has never been documented, and it is extremely rare among non-*brasiliensis Sporothrix*. However, transmission is possible when close daily interactions involve direct contact with lesions [39,40]. The infrequency of these infections suggests that humans do not play a major role in the natural history of *Sporothrix* spp. and transmission is exclusively by zoonotic and environmental routes.

Cats are the primary animal host for *S. brasiliensis* and vector of infection [41]. In the Rio de Janeiro sporotrichosis epidemic, ill cats typically had a high fungal burden on skin lesions and were the main source of *S. brasiliensis* infection for other cats, dogs, and humans [9,42]. Previous studies in Rio de Janeiro have found that as many as 87% of humans and 78–84% of dogs with sporotrichosis had documented exposure to cats before becoming ill [31,32,33].

Rats (*Rattus norvegicus*) may also play a role in the propagation of *S. brasiliensis* among cats, which could potentiate outbreaks among people, although evidence is limited. The isolation of *Sporothrix* from naturally infected rats demonstrated their susceptibility to sporotrichosis, and documented cases of human sporotrichosis have been reported following rat bites in Argentina and Brazil [12,20]. Given frequent interactions between cats and rats in densely populated urban environments, further exploration of *S. brasiliensis* in rats is warranted. Additionally, squirrels, armadillos, fish, cockatiel, and insects have been identified as possible mechanical vectors for zoonotic transmission caused by *Sporothrix* spp., but there has been no evidence that they are sources of infection for sporotrichosis caused by *S. brasiliensis* [43,44,45,46,47,48].

Dogs are not a major source for human infections of *S. brasiliensis*. Outbreaks of *S. brasiliensis* have occurred among dogs in Brazil, but they carry a low burden of yeast cells in their lesions [9,35]. There have been few documented dog-to-human cases of sporotrichosis [42,49], but a study of dogs with sporotrichosis found that *S. schenckii* species complex could be isolated from the oral cavity and conjunctival mucosa, confirming the biological plausibility of dog-to-human transmission [50].

In addition to the animal reservoirs for *S. brasiliensis*, this species has occasionally been identified from environmental sources. Environmental isolation is uncommon, however, in comparison to other *Sporothrix* spp., which have been associated with soil, plants, woody debris, and corn stalks [51,52,53]. For example, *S. brasiliensis* was isolated from a soil sample taken from the cave of a nine-banded armadillo (*Dasypus novemcinctus*) in Argentina in 2003 [12]. Another study isolated *S. brasiliensis* from the small intestinal contents of two necropsied cats, as well as from cat feces collected from a pile of sand in São Paulo [35]. Feces from infected cats may contaminate the soil, creating an environmental reservoir for *S. brasiliensis* and a possible new source of contamination for animals or humans [35]. While no documented cases of sapronotic transmission of *S. brasiliensis* exist, the presence of fungi on plant and organic matter makes this type of transmission plausible.

## 6. Human Sporotrichosis

### 6.1. Transmission

Transmission of pathogenic *Sporothrix* species other than *S. brasiliensis* occurs most commonly by traumatic inoculation of fungus found on plant matter; however, this route has never been documented for *S. brasiliensis*. Nearly all reported human infections with *S. brasiliensis* are caused by traumatic inoculation via the bite, scratch, or contact with exudate from the cutaneous lesions of ill cats [9,10,11,54]. Non-traumatic routes of infection with *S. brasiliensis* may include direct contact with sporotrichosis lesions, droplet exposure on mucous membranes, and inhalation of mold and potentially yeast forms [41,52,55]. A study of 33 cats found that the lesions of 29 cats had a high burden of yeast (>25 cells/HPF), which facilitates transmission via traumatic and non-traumatic routes [17]. In contrast, a study of 44 dogs with sporotrichosis found that only six had detectable yeast-like elements by direct microscopy, demonstrating a low fungal load and low likelihood for transmission [34]. Unlike other dimorphic fungi, which only cause infection when transmitted by the mold form, *S. brasiliensis* is unique in its ability to be transmitted in the yeast phase as well, a key factor in facilitating transmission between mammals [7,9]. Notably, groups at high risk for CTS include those who have frequent contact with cats, such as veterinarians, veterinary technicians, animal caretakers, and cat owners [3].

### 6.2. Clinical Presentation

In humans, *S. brasiliensis* and non-*brasiliensis Sporothrix* infections present most commonly as cutaneous or lymphocutaneous lesions, including papules, nodules, and ulcers (Figure 1). Dr. Flavio Queiroz-Telles provided informed consent for the use of these images. These lesions often appear at the site of trauma a few days to several months after exposure [56,57]. Cutaneous lesions may rapidly spread through regional lymphatic vessels, resulting in lymphatic nodules and abscesses. Lymphocutaneous lesions are seen in 95% of sporotrichosis cases, whereas disseminated forms occur in <10% of cases [52,58,59].

In addition to cutaneous presentations, *S. brasiliensis* is less commonly associated with ocular involvement, disseminated disease, central nervous system (CNS) disease, and hypersensitivity reactions [11,60,61,62,63,64]. The most common ophthalmic manifestations are acute and chronic conjunctivitis, dacryocystitis, and Parinaud syndrome, although cutaneous eyelid infections and other ocular sequelae can also occur [4,65]. Ophthalmic infections can result when secretions from an infected cat contact a person’s conjunctiva [65]. Hypersensitivity reactions are a well-documented, though less common, manifestation of CTS and are uniquely associated with feline-to-human transmission. These reactions do not seem to occur in infections with non-*brasiliensis* species. They result from a cell-mediated immune response to *S. brasiliensis* antigens, but the exact immune mechanism is unknown. In one study, 10 of 45 cases of sporotrichosis caused by *S. brasiliensis* involved a hypersensitivity reaction, whereas zero of five cases of *S. schenckii* had this finding [11]. The most common manifestations include erythema nodosum and multiforme, arthralgia, myalgia, and arthritis. These clinical manifestations can be similar to dermatophytid reactions, which are inflammatory reactions to a fungal infection at a distant body site [11,64,66,67]. The least common CTS presentation is disseminated infection, which can include osteoarticular, pulmonary, and neurologic infections [4,11]. Underlying factors associated with disseminated sporotrichosis include immunosuppressive conditions, alcoholism, and diabetes [7].

### 6.3. Treatment

Although *S. brasiliensis* is more virulent than other *Sporothrix* species in animal models and shows less in vitro sensitivity to some antifungal drugs, human sporotrichosis treatment is the same regardless of the infecting species [6,68]. According to the Infectious Diseases Society of America, the first line therapy for fixed cutaneous, lymphocutaneous, and osteoarticular cases of human sporotrichosis is itraconazole (ITZ) at 200–400 mg/day orally [69]. A more recent report on treating CTS specifically found that therapeutic response rates were high with a dose of 100 mg/day [66]. Until the 1990s, saturated solution of potassium iodide (SSKI) was used to treat cutaneous and lymphocutaneous clinical forms, but SSKI is not well tolerated and can cause thyroid dysfunction. However, SSKI remains a treatment option in low resource regions, where ITZ or terbinafine are not available [70,71]. Alternatively, terbinafine at 500 mg twice daily may be used for fixed cutaneous and lymphocutaneous forms if ITZ is contraindicated [72]. Additionally, cryosurgery and local heat therapy may be used in conjunction with antifungals to reduce therapy duration, for the treatment of pregnant women, or to treat recalcitrant, hyperkeratotic lesions [73,74,75,76,77].

In the case of severe pulmonary, CNS, or disseminated sporotrichosis, lipid-formulation amphotericin B at 3–5 mg/kg/day is preferred [69]. The duration of antifungal regimens range by clinical form, from two to six months, and may be extended beyond 12 months for severely ill patients and when patients fail to respond to therapy [69,70].

## 7. Feline Sporotrichosis

### 7.1. Feline Epidemiology

Similar to humans, cats can become infected with *Sporothrix* spp. by two routes: inoculation of the fungus found on decaying plants, soil, or organic matter, or by scratches, bites, or contact with fluids from infected cats [8]. While other *Sporothrix* spp. have extensive sapronotic transmission, evidence suggests that *S. brasiliensis* transmission occurs almost exclusively through cat-to-cat contact. In one study, 93% of cats with sporotrichosis had a history of contact with other cats [78]. A study by de Souza et al. found that 7 (29.1%) of 24 healthy cats living with a cat with clinical sporotrichosis tested positive for *S. schenckii* by culturing their paws, showing that asymptomatic carriage and transmission of *Sporothrix* spp. from healthy cats is possible [79]. However, it seems that healthy cats play a minor role in *Sporothrix* transmission, as there is a low frequency of fungal isolation from the oral cavity and claws of healthy cats that had contact with infected cats [3].

*Sporothrix brasiliensis* is the primary cause of feline sporotrichosis in Brazil, and while information on the epidemiology of feline sporotrichosis is limited, disease occurs most frequently among adult male, mixed-breed, and unneutered cats [1,35,80,81,82,83,84]. Although human cases of disseminated sporotrichosis occur more commonly among people with immunosuppressive conditions [7], a similar association has not been noted for feline sporotrichosis. Surprisingly, co-infections with Feline Immunodeficiency Virus (FIV) or Feline Leukemia Virus (FeLV) has not been associated with increased incidence of sporotrichosis, more severe clinical or laboratory findings, nor worse clinical outcomes in most studies [78,85,86]. However, one study found that cats co-infected with these retroviruses had significant differences in cytokine transcription levels (higher IL-10 and lower IL-4, IL-12, and CD4+/CD8+ ratio) when compared to other cats with sporotrichosis and no retroviral co-infection. All of the cats in this study identified to have a poor general health condition or severe clinical manifestations were also found to have a retroviral co-infection [87]. The small sample size of cats with FELV and FIV co-infections in previous sporotrichosis studies may have resulted in insufficient power to identify potential differences among this subgroup, so further investigation is warranted.

Cats with severe sporotrichosis and negative for FIV and FeLV may have other comorbidities or immunosuppressive conditions [87]. Parasitic co-infections with tapeworms (*Taenia taeniaeformis*) among rats with *S. schenckii* were associated with immunological changes that reduce the host’s ability to combat fungal infections, so the role of helminth co-infections in cats with sporotrichosis should also be investigated [88].

### 7.2. Clinical Presentation

The most common clinical manifestations include multiple ulcerated skin lesions associated with enlarged lymph nodes and the presence of respiratory signs (mainly sneeze) [81,89]. The incubation period is similar to that described for human infections, with onset typically occurring within three to 30 days after exposure, though it may extend for months [89,90]. The skin lesions are characterized by nodules and ulcers found in different anatomical sites, commonly on the head (especially in the nasal region) and limbs [78,85]. Lymphangitis and mucosal lesions have also been observed (most commonly on the nasal mucosa) [89]. Respiratory signs and nasal mucosal lesions are significantly associated with therapeutic failure [83,85] (Figure 2).

Disseminated disease also occurs, and cats with a history of lethargy, depression, anorexia, and fever are more likely to develop disseminated disease [91]. In one study, isolation of *Sporothrix* spp. from peripheral blood of cats with early infection and localized or multiple skin lesions showed that hematogenous spread of the fungus can occur soon after infection [86]. Other signs of disseminated disease include anemia, hyperglobulinemia, hypoalbuminemia, and leukocytosis with neutrophilia, and these changes are more frequent among cats presenting multiple skin lesions [78].

### 7.3. Treatment

The treatment of feline sporotrichosis can be challenging. Early diagnosis, rapid establishment of a therapeutic regimen, and the owner’s cooperation are critical to achieve clinical cure. Itraconazole (ITZ) and potassium iodide (KI) are the most commonly used drugs for the treatment of feline sporotrichosis, regardless of clinical form [85,92]. ITZ is the preferred drug at a dose of 8.3–27.0 mg/kg/day orally, and it is associated with less adverse reactions than ketoconazole [42,78,83,85,93,94,95]. KI capsules are favored in some regions, including Brazil, because it costs less than ITZ and represents an important option for cases refractory to ITZ monotherapy in all regions where sporotrichosis is seen [92,96]. One study found that use of ITZ capsules (100 mg/day orally) in conjunction with KI capsules (2.5–20 mg/kg/day orally) could be considered for treatment-naïve cats, as it was shown to have a faster onset of action than ITZ or KI monotherapy. The adverse reactions of ITZ and KI are similar to those observed in ITZ alone [85,97]. These reactions are managed with a temporary drug suspension and an appropriate clinical follow-up [97]. Overall, *S. brasiliensis* presents good activity in vitro to antifungals [98,99,100]. Further information on therapeutic regimens for feline sporotrichosis can be found in Appendix B.

It is important to note that therapeutic failure and recrudescence are common in feline sporotrichosis, especially among animals with nasal, skin, and mucosal lesions or respiratory signs [17,85]. Cats presenting with multiple skin lesions tend to have persistent lesions and a higher rate of ITZ monotherapy treatment failure [87]. In addition, high fungal burdens observed in skin lesions before treatment with ITZ were associated with treatment failure and longer time to clinical resolution [83].

The criterion for cure of feline sporotrichosis is clinical, with the disappearance of all clinical signs [17]. The reactivation of lesions following treatment resolutions has been well documented, so adherence to a treatment protocol that extends at least one month beyond clinical cure is crucial [101,102].

## 8. Diagnosis of Human and Feline Sporotrichosis

Sporotrichosis diagnosis in humans, cats, and other mammals can be performed using several common laboratory methodologies, including fungal culture, direct examination, immunodiagnostic methods, molecular assays, and proteomic approaches (Table 1). Each of these methods will be discussed in more detail, along with factors that may affect the performance of these laboratory assays and decisions about which to use [103,104]. Lastly, the use of antifungal susceptibility testing (AFST) for *Sporothrix* will be briefly discussed.

Fungal culture is the reference standard technique for diagnosing sporotrichosis caused by any species. *Sporothrix* spp. grow in 5 to 8 days on a variety of fungal media. The filamentous form is recovered using Sabouraud dextrose at 28 °C. Macroscopically, filaments are apparent, with colonies that are moist with a finely wrinkled surface and initially appear white but become dark brown [104,105]. Microscopically, hyphae are septate and conidia are oval-shaped and resemble a flower. Growth of *Sporothrix* spp. in the yeast form requires culturing at 37 °C on Blood Heart Infusion, blood, or blood-chocolate agars [106]. Colonies appear smooth, white or off-white, and are comprised of elongated cigar-shaped yeast cells that are 2–6 μm long [104,105]. For conventional identification of clinically relevant *Sporothrix* spp. strains when molecular testing is not available, Rodrigues et al. proposed a dichotomous key, based on phenotypic features (morphology of conidia, growth at different temperatures, and carbohydrate assimilation) [19,107].

Direct examination of fungal structures is a microscopic technique to visualize yeast in cytological preparations and biopsy specimens, and it can be used for the diagnosis of human and animal sporotrichosis. *Sporothrix* spp. can be identified or isolated from skin lesions, respiratory specimens, synovial fluid, and blood [104,105]. Direct examination involves a slide preparation of one of the above specimens with a 10–40% KOH solution to facilitate visualization. A positive direct test is characterized by the presence of yeasts of 2–6 µm in diameter, but the yeasts are difficult to observe in human samples [104,105,108]. Periodic acid-Schiff and Gomori methenamine silver fungal stains should be used to better visualize fungal structures, but even with staining, direct examination of human samples has low sensitivity [104,105,108]. In contrast, samples from infected cats have a much higher fungal load, with positive identification often possible at 400× magnification, and sensitivity ranging from 79% to 87% at 1000× magnification [52,109,110]. This technique is often used for the diagnosis of feline sporotrichosis due to its low cost, easy execution, and rapid results, and it does not require sophisticated technical training or complex laboratory structure [109,111]. Lastly, in humans, a special preparation of samples from lesions on a slide with a drop of physiological saline solution and a drop of 10% formaldehyde solution can be used to directly observe asteroid bodies, yeast cells surrounded by a radiating pattern of eosinophilic immunoglobulins. The sensitivity of this technique is >90%, and asteroid bodies can be observed in ~43% of sporotrichosis cases [108]. Importantly, a negative result by direct examination does not exclude fungal infection, and therefore, fungal culture results should be used for confirmation. In cats, direct examination can also be used to monitor the fungal burden of ulcerated skin lesions during antifungal treatment, and to help inform dosage and duration of therapy [42].

Immunodiagnostic methods remain experimental and are not widely used. A latex agglutination test system for detection of specific anti-*Sporothrix* spp. antibodies is commercially available (IMMY©, Norma, OK, USA) [112]. This test is offered for the presumptive diagnosis of sporotrichosis. This is a highly specific test for the genus *Sporothrix* (100%), but sensitivity varies according to clinical presentation in humans: 100% for disseminated forms, 86% for osteoarticular forms, 73% for pulmonary, and 56% for cutaneous forms.

Molecular assays based on DNA sequence identification are promising as an alternative diagnostic method. Multiple publications describe various molecular protocols including real-time PCR, calmodulin sequencing, rolling circle amplification, PCR-RFLP, and others. These assays were evaluated using different *Sporothrix* species, isolates, and types of specimens, and in general, most of the protocols target the calmodulin gene (considered the reference standard for species-level identification) or nuclear ribosomal internal transcribed spacer (ITS) region. Sensitivity and specificity were consistently high across assays (>90%) [113,114,115,116,117,118,119,120,121]. However, molecular assays require advanced laboratory infrastructure, and utility is limited by the lack of commercially available kits, standardized methods, and a limited number of validation studies.

A proteomic approach using matrix-assisted laser desorption ionization time-of-flight mass spectrometry (MALDI-TOF MS) has also been used to accurately identify *Sporothrix* species. A study from Brazil used 70 environmental and clinical isolates of *Sporothrix* spp. in the pathogenic clade and tested them using MALDI-TOF MS. The test was able to distinguish all isolates at the species level, including *S. brasiliensis*, *S. globosa, S. mexicana, S. schenckii, S. luriei,* and *S. pallida*. Isolate identification in this study was confirmed by sequencing of the calmodulin gene [122].

Antifungal susceptibility testing conditions for the filamentous form of *Sporothrix* spp. have been described by the Clinical and Laboratory Standards Institute (CLSI M38 reference method). However, formal breakpoints (BPs) have not been described for *Sporothrix* spp., though a multicenter study proposed epidemiological cutoff values (ECVs) for some species [100,123].

## 9. Prevention and Control

*Sporothrix brasiliensis* is now the leading causative agent of human and feline sporotrichosis in Brazil, and zoonotic transmission poses unique challenges in controlling and preventing this disease. Because infected cats are the main source of *S. brasiliensis* human and animal infections, control and prevention efforts focus primarily on reducing the burden in the feline population to reduce the risk of cat-to-cat, cat-to-other animal, and cat-to-human transmission.

### 9.1. Preventing Transmission Among Cat Populations

To prevent the spread of CTS from feral cat populations into domestic cats, pet owners should prevent their domestic cats from interacting with feral cats by restricting their feline companions’ access to the outdoors.

Additional activities that could reduce the risk of infection and carriage of *S. brasiliensis* in feline populations include reproductive control and partnering with feral cat programs. Through spay and neuter campaigns, it would be possible to reduce the number of intact, male cats, which are overrepresented among feline sporotrichosis cases [17,78,81,84]. Partnering with feral cat programs, such as vaccine campaigns, would allow collaboration between public health and veterinarians, sharing of resources, and capture of potentially infected cats for early intervention.

Addressing active infections among the cat population is important to control the spread of CTS. Infected cats should be treated with antifungal medication for at least one month beyond clinical cure. Studies have shown that some therapeutic protocols may reduce the fungal burden in cats with sporotrichosis caused by *S. brasiliensis*, promoting the early treatment of feline sporotrichosis as a control measure [42,83]. An effective vaccine for human or feline sporotrichosis does not yet exist, but development of such a vaccine could be instrumental in the control of CTS [124]. If a cat cannot be treated for sporotrichosis, euthanasia can prevent further dissemination of *Sporothrix* into the feline population or environment. Additionally, cremation is recommended in lieu of backyard burial to prevent possible environmental contamination [14].

### 9.2. Preventing Cat-to-Human Transmission

Veterinarians, veterinary technicians, animal caretakers, and cat owners (especially those who own multiple cats with unrestricted outdoor access) are at greatest risk of contact with infected cats and should take additional precautions to prevent CTS [3,125]. To prevent transmission, when handling sick cats, gloves, masks, and eye protection should be worn to avoid transmission via traumatic inoculation, inhalation of fungal spores, or conjunctival contact with infectious droplets [126].

Members of the general public can also take precautions to prevent exposure to *S. brasiliensis*. People should be cautious with unfamiliar animals and approach cats with care, even if they appear friendly. Avoiding bites and scratches will help to prevent *S. brasiliensis* infection, thus stopping the primary mode of zoonotic transmission [127]. If a person is bitten or scratched by a cat, they should immediately wash the wound with soap and warm water and seek medical attention, as cat bites and scratches can lead to infection with a variety of pathogens [127].

Lastly, precautions should be taken at ports of entry to ensure that *S. brasiliensis*-infected animals are not relocated to non-endemic areas. Owners relocating with their pets should have the proper health certificates, and pets should be visually inspected for skin and mucosal lesions upon entry. Travel-associated CTS has been described, involving an infected cat that was transported from Brazil to a non-endemic country (Paraguay) and a person who developed zoonotic sporotrichosis caused by *S. schenckii* after contact with an infected cat in Minas Gerais, Brazil (Boston, USA), so this potential source of spread should not be ignored [13,38]. Cat owners moving to endemic areas should be warned about sporotrichosis and should restrict their pet’s access to outdoor areas [128].

## 10. A One Health Approach to Cat-Transmitted Sporotrichosis Outbreak

One Health is an approach to achieving optimal health outcomes through interdisciplinary collaborative efforts that focus on the interrelationships among humans, animals, plants, and the environment. Combatting the spread of *S. brasiliensis* with a One Health approach will require multisectoral efforts that include veterinarians, physicians, epidemiologists, microbiologists, environmental scientists, and many other partners.

The use of a One Health approach could improve our understanding of the epidemiology of *S. brasiliensis* and help combat further spread. The veterinary discipline has the opportunity to fill critical roles in investigating and mitigating these outbreaks. Veterinarians will be needed to address the spread of *S. brasiliensis* among the feline population, as well as investigating other potential host species. Their roles should include the treatment of infected animals, education of pet owners on the potential for zoonotic exposure, and the promotion of appropriate husbandry and mitigation efforts necessary to limit the spread of *S. brasiliensis*. When educating clients and discussing treating feline sporotrichosis, it is important for veterinarians to highlight the difficulties of treatment, including the need for long-term adherence to an antifungal regimen, risk of adverse reactions to these medications, and the need to keep infected cats indoors to prevent spread of this fungus during treatment. Additionally, when owners decline treatment or when treatment fails, veterinarians will need to discuss euthanasia and cremation, as releasing infected cats or burying the remains of their pet could introduce fungal spores into the environment and perpetuate spread.

Veterinarians are also integral in their role as providers of International Health Certificates, which allow for the movement of animals between countries. When providing these health certificates, it is important to conduct a thorough physical exam, perform any necessary testing to rule out infectious diseases, and ensure that the proper, country-specific health certificate requirements are met.

In the setting of a CTS outbreak, a collaborative, multidisciplinary approach could be crucial. To learn more about the epidemiology of *S. brasiliensis*, the testing of cats and other animals should be considered to determine both the burden of disease in the feline population, as well as detect any other species that may serve as reservoirs for this pathogen. Collaboration with environmental scientists and microbiologists is essential for collecting, storing, shipping, and testing of environmental and clinical samples. Simultaneously, physicians should be alert for cases of CTS, request the proper diagnostics to promptly diagnose sporotrichosis, and ensure rapid initiation of antifungal therapy. Physicians and veterinarians should also collaborate on the investigation of suspected zoonotic outbreaks, development and enhancement of surveillance systems for human and feline sporotrichosis, and research into One Health techniques for controlling CTS.

Throughout these outbreak investigations, epidemiologists will be needed to design studies to better understand the epidemiology of *S. brasiliensis*, track trends in feline and human cases of sporotrichosis, analyze the data collected by the various partners, determine any risk or protective factors among these species, and ensure that a comprehensive and uniform surveillance system is in place. To fully understand the epidemiology of *S. brasiliensis* and implement holistic public health control measures, a One Health approach that engages diverse disciplines is essential.

## 11. Conclusions

*Sporothrix brasiliensis* has led to a shift in the traditional epidemiology of sporotrichosis, with a drastic increase in CTS, and the zoonotic nature of this emerging pathogen poses unique barriers to halting its spread. This etiologic agent originated in Southern Brazil but has spread widely in the past 20 years, resulting in an endemic region that now includes Argentina. With its spread beyond Brazil and ongoing risk of introduction to other countries, there is an urgent need to learn more about this important fungal pathogen.

Some of the major barriers to controlling CTS include the lack of systematic surveillance, availability of rapid diagnostic tests, adequate treatment and control of disease in the feline population, and limited awareness of this disease in non-endemic countries. The development of widespread systematic surveillance in endemic countries is needed to understand the scope and geographic reach of feline sporotrichosis and CTS. Cats can move unimpeded across state and national borders, which poses a risk for introductions of *S. brasiliensis*. Use of consistent surveillance methods will allow states and countries to monitor trends in incidence and geographic spread that could help direct public health resources to prevent further spread.

The lack of commercially available diagnostic tests for early detection of *S. brasiliensis* is a limitation to preventing the spread of this pathogen to naïve populations. To facilitate the rapid diagnosis of *S. brasiliensis* and enhance surveillance efforts in endemic and neighboring countries, increased laboratory capacity for molecular testing is needed beyond a select few reference laboratories. In addition, commercially available molecular diagnostic kits and inclusion of *Sporothrix* spp. spectrum in MALDI-TOF databases capable of distinguishing *S. brasiliensis* from other *Sporothrix* species would improve diagnostic performance and reduce turnaround time compared with current standard methods. In the meantime, physicians and veterinarians should consider submitting samples to reference laboratories for molecular testing to determine the infecting species when they have a case with presumptive diagnosis of sporotrichosis and an epidemiologic link to cats. Addressing these barriers can help prevent these serious infections in cats and in humans.

Feline sporotrichosis spreads widely within cat populations, which presents a hurdle to controlling zoonotic transmission. Because many cats have unrestricted outdoor access, they are likely to infect other cats and shed the fungus into the environment for an extended period if they become infected. Veterinary initiatives that address the health of stray and feral cat populations—including animal sterilization programs and vaccination campaigns—should be leveraged as opportunities pool resources to improve the welfare of outdoor cats, while helping to understand the burden of *S. brasiliensis* in the feline population and allowing for early interventions.

Lastly, limited awareness of this emerging pathogen among physicians and veterinarians also threatens containment efforts. In countries that have never had a case of *S. brasiliensis*-associated CTS, as well as countries with diagnosed autochthonous cases, limited familiarity with this emerging pathogen could lead to delays in proper diagnosis, poor outcomes, and the potential for further spread of disease. Education campaigns should aim to provide information to veterinary and human medical providers, focusing on the interconnectedness of the human and feline control strategies, available diagnostic options, and proper clinical management of disease.

The increased virulence, treatment challenges, and adaptation to the feline host make *S. brasiliensis* especially challenging to control, and the impact of this pathogen on human and feline populations has been substantial. The risk of continued outbreaks in Brazil and spread into other countries further underscores the importance of addressing this emerging disease. Taking a One Health approach to combatting *S. brasiliensis* could help address the barriers to adequate control, shed light on the epidemiology of this emerging fungus, and allow for the implementation of holistic public health actions to prevent the spread of sporotrichosis.

## Figures and Tables

**Figure 1 jof-06-00247-f001:**
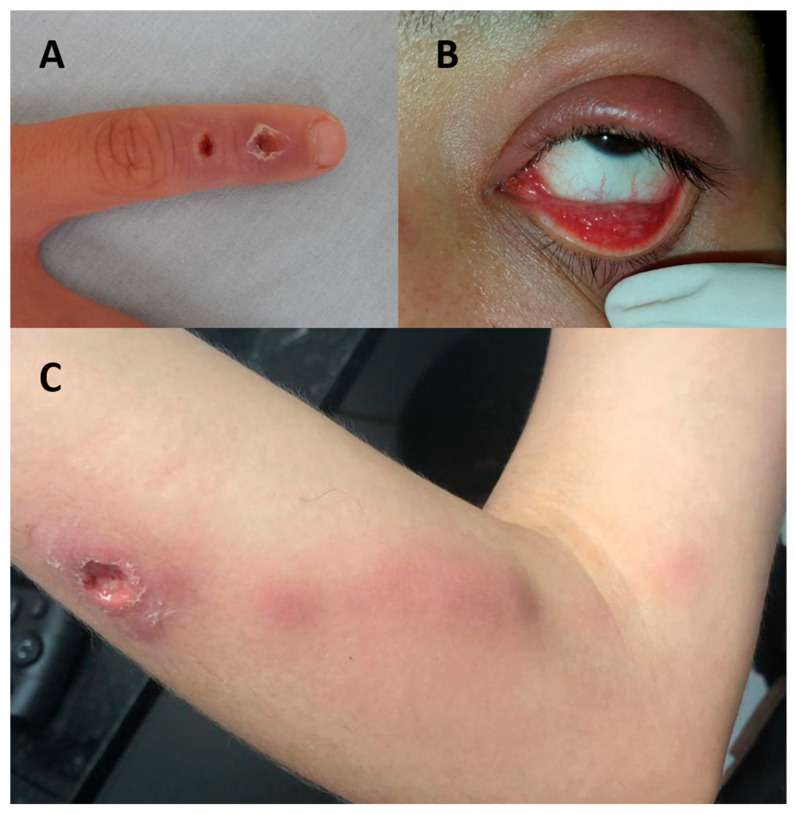
Clinical presentations of cat-transmitted sporotrichosis in humans. (**A**) Fixed cutaneous presentation: isolated healing ulcers at the site of trauma. (**B**) Ophthalmic manifestation: granulomatous conjunctivitis. (**C**) Lymphocutaneous presentation: a large ulcerated lesion at the site of trauma with evidence of lymphatic nodules and erythema along the regional lymphatic vessel. Photo credit to Dr. Flavio Queiroz-Telles.

**Figure 2 jof-06-00247-f002:**
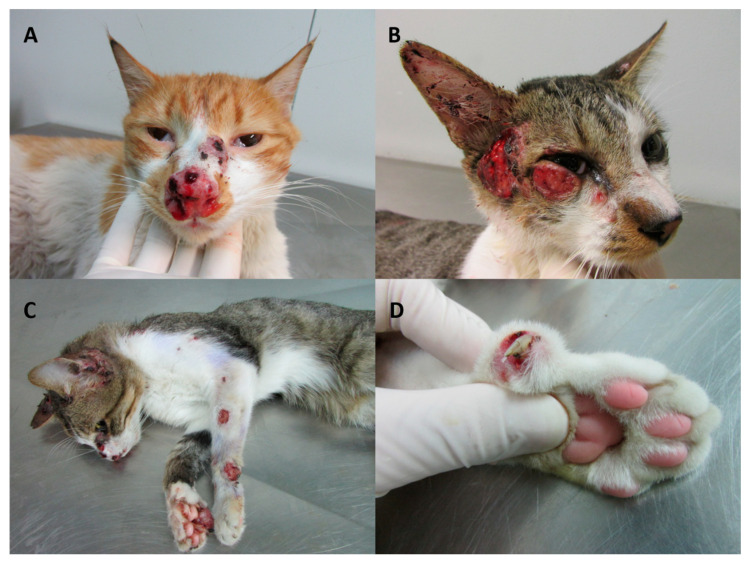
Clinical presentations of feline sporotrichosis. (**A**) Swelling of the nasal region with crusted skin ulcers on the nasal bridge and nasal planum, draining serosanguinous exudate. Serosanguinous nasal discharge. (**B**) Ulcerated skin lesions and crusts on the cephalic region. Epiphora. (**C**) Multiple skin lesions on the cephalic region. Ascending nodular lymphangitis on the left forelimb and tumor-like lesion on the upright paw. (**D**) Paronychia on the forelimb. Photo credit to the Laboratory of Clinical Research on Dermatozoonoses in Domestic Animals (Lapclin-Dermzoo), Evandro Chagas National Institute of Infectious Diseases (INI), Oswaldo Cruz Foundation (Fiocruz). Use of these images was authorized by Lapclin-Dermzoo/INI/Fiocruz.

**Table 1 jof-06-00247-t001:** Laboratory approaches for the diagnosis and treatment of sporotrichosis.

Technique	Specimen Type	Characteristics
Culture	Affected tissue, body fluids, or blood	5–8 days to grow in various fungal media (e.g., Sabouraud dextrose agar, Mycosel agar, Brain Heart Infusion agar) 28 °C: Filamentous form with septate hyphae, oval conidia (resembles a flower) 37 °C: Yeast form with 2–6 μm cigar-shaped yeast cells
Microscopy	Cytological preparation or lesion biopsy	Direct test (10–40% KOH): 2–6 µm yeasts, rarely seen (1–2% of cases) PAS and GMS: oval or cigar-shaped organism 10% formaldehyde (1 drop) and physiologic saline (1 drop): observation of asteroid bodies In cats: Romanowsky-type stains (e.g., Quick Panoptic, Diff-Quick or Wright) or Gram staining
Immuno-diagnosis	Serum	Commercially available latex agglutination test system for antibody detection, sensitivity varies with clinical form (data from humans): ▪ 100% for disseminated sporotrichosis ▪ 86% for osteoarticular sporotrichosis ▪ 73% for pulmonary sporotrichosis ▪ 56% for cutaneous sporotrichosis Other assays remain experimental
DNA Detection	Affected tissue, fresh or formalin-fixed paraffin-embedded tissue, body fluids, or *Sporothrix* isolate	Most DNA detection assays target the calmodulin gene DNA sequences represent the nuclear ribosomal internal transcribed spacer
MALDI-TOF MS	*Sporothrix* isolate	Able to distinguish to the species level (*S. brasiliensis*, *S. globosa*, *S. mexicana*, *S. schenckii*, *S. luriei*, and *S. pallida*).
Antifungal Susceptibility Testing	*Sporothrix* isolate	Mold CLSI reference method (M38), does not discriminate by *Sporothrix* species. A multicenter study proposed the following epidemiological cutoff values for minimal inhibitory concentration/minimal effective concentration: *S. schenckii* and *S. brasiliensis*: amphotericin B: 4 μg/mL; itraconazole: 2 μg/mL; posaconazole: 2 μg/mL; voriconazole: 64 μg/mL for *S. schenckii* and 32 μg/mL for *S. brasiliensis*. *S. brasiliensis*: ketoconazole 2 μg/mL and terbinafine 0.12 μg/mL.

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
