# Peer review of "A One Health Approach to Combatting Sporothrix brasiliensis: Narrative Review of an Emerging Zoonotic Fungal Pathogen in South America"

_jof, 2020, doi:10.3390/jof6040247_

Round 1

Reviewer 1 Report

This narrative review put together relevant information on zoonotic sporotrichosis. In the past few years, inumerous studies were carried out in an attempt to resolve gaps in this important zoonosis and a lot of knowledge has been built within this time. In this sense, this review can be extremely useful to wrap up these new findings, providing a summary of what we know so far. Please, see below some relevant comments and inputs regarding the manuscript presentation.

Page 1, Lines 38-39: Environmental transmitted sporotrichosis is probably more widely distributed, but the total number of cat-acquired sporotrichosis exceeded the environmental cases, greatly due to the Brazilian epidemics. Cat-transmitted sporotrichosis seems to have a longer reach with probably a higher number of people exposed and a source of infection that is difficult to control. The authors may want to check these numbers and rewrite the sentence accordingly.

Throughout the manuscript, favour the use of species full name at the beginning of a sentence or paragraph, and review the species name that are not in italic.

Page 2, Lines 52-53: It might be a bit too strong to state this species possibly expanded to US based on the references provided by the authors. In fact, there is a case described in a Brazilian woman that travelled to US from Brazil, however, the causative species isolated was not confirmed as S. brasiliensis. Still, although the carriage of a new pathogen species is a noticeable event, human does not seem to have a strong epidemiological role in sporotrichosis as they do not serve as source of infection from what we know so far. Having said that, I do encourage the authors to keep the mention to this recent report, but to appraise this information more critically.

Material and Methods: The organization of this section would improve in clarity if the authors kept the topics “causal agent”, “geographic distribution” and “Animal hosts, reservoirs, and sources of human S. brasiliensis infection” as separated topics (comprising aspects from both human and animal sporotrichosis), rather than embedded on the topic “Human sporotrichosis”. The scope of this manuscript is the ‘one health’ approach, so it is just as well that a comprehensive presentation of the information is brought, as human and feline sporotrichosis due to S. brasiliensis are substantially related.

Page 3, Lines 108-111: The identification of S. brasiliensis in environmental samples from the 80’s is a important finding and points out to the fact that frank zoonotic spread was described only longer after that in Brazil. This should be discussed in a broader way as a shift from environment to mammal host is usually debated in Sporothrix species dynamic as a crucial event to the zoonotic turnover. This is absolutely a keypoint to reinforce the relevance of this pathogen to the ‘one health’ discussion.

Page 3, Lines 24-25: This sentence should be more assertive, not only suggesting, but stating that “cats are the primary animal host for S. brasiliensis and vector of infection”. The same applies for line 138, that suggests dogs are not a major source of S. brasiliensis to humans. Even if cases of dog to human transmission went unnoticed for any reason, they definitely do not have a significant epidemiological role in this scenario.

Page 4, Lines 163-164: The sensitivity for detecting Sporothrix in lesions of canine sporotrichosis is generally low, but it’s reported as high as 40%. Still, the fungal load is so low that makes the transmission unlikely.

Page 6, Line 247: Not only Tapeworm, I would suggest helminthic diseases in general, as anti-helminthic  immunity may antagonize or suppress the immune response to other pathogens.

Page 8, Line 298: Please, provide more information on the adequate specimen for each kind of diagnostic method within the text or embedded in the table.

Page 8, Lines 316-334: This paragraph needs more clarification and clearly state that yeast cells can be detected either from cytological preparations or biopsy specimens. Asteroid bodies are not common in lesions of human sporotrichosis from the Brazilian outbreaks and this issue is not really on the scope of the reference provided unless there was some referencing mistake.

Page 10, Lines 356-359: Please, consider including antifungal susceptibility assays into the treatment component.

Page 11, Line 383: This section is lacking some references, especially on the first and second paragraphs  

Page 12, Lines 482-483: I don’t think the ability to persist in the natural environment was an explored feature in this paper, as I don’t see this as a particularity of S. brasiliensis. Sporothrix brasiliensis, as pointed in this paper, seems much more adapted and hard to control in the feline host.

Author Response

Reviewer 1 Comments:

Page 1, Lines 38-39: Environmental transmitted sporotrichosis is probably more widely distributed, but the total number of cat-acquired sporotrichosis exceeded the environmental cases, greatly due to the Brazilian epidemics. Cat-transmitted sporotrichosis seems to have a longer reach with probably a higher number of people exposed and a source of infection that is difficult to control. The authors may want to check these numbers and rewrite the sentence accordingly.

R. The sentence starting on Line 38 has been revised to highlight that sapronotic transmission was the most common source of sporotrichosis historically, but with the emergence of brasiliensis, this has shifted to zoonotic transmission.

Throughout the manuscript, favour the use of species full name at the beginning of a sentence or paragraph, and review the species name that are not in italic.

R. The species names were reviewed and italicized where needed throughout the manuscript. The genus has also been spelled out when used at the beginning of a sentence or paragraph.

Page 2, Lines 52-53: It might be a bit too strong to state this species possibly expanded to US based on the references provided by the authors. In fact, there is a case described in a Brazilian woman that travelled to US from Brazil, however, the causative species isolated was not confirmed as S. brasiliensis. Still, although the carriage of a new pathogen species is a noticeable event, human does not seem to have a strong epidemiological role in sporotrichosis as they do not serve as source of infection from what we know so far. Having said that, I do encourage the authors to keep the mention to this recent report, but to appraise this information more critically.

R. The mention of a potential case in the United States has been removed from the sentence at Line 54, but this recent report is mentioned in a new sentence on Page 3, Line 121 to highlight that the carriage of a pathogen to a naïve country is possible.

Material and Methods: The organization of this section would improve in clarity if the authors kept the topics “causal agent”, “geographic distribution” and “Animal hosts, reservoirs, and sources of human S. brasiliensis infection” as separated topics (comprising aspects from both human and animal sporotrichosis), rather than embedded on the topic “Human sporotrichosis”. The scope of this manuscript is the ‘one health’ approach, so it is just as well that a comprehensive presentation of the information is brought, as human and feline sporotrichosis due to S. brasiliensis are substantially related.

R. The topics on “Causal Agent”, “Geographic Distribution”, and “Animal hosts…,” have been separated from the “Human Sporotrichosis” section to help present the information more clearly, as suggested.

Page 3, Lines 108-111: The identification of S. brasiliensis in environmental samples from the 80’s is a important finding and points out to the fact that frank zoonotic spread was described only longer after that in Brazil. This should be discussed in a broader way as a shift from environment to mammal host is usually debated in Sporothrix species dynamic as a crucial event to the zoonotic turnover. This is absolutely a keypoint to reinforce the relevance of this pathogen to the ‘one health’ discussion.

R. On Page 3, Line 112, a sentence has been added to reinforce the importance of a One Health approach to combatting this pathogen, as well as highlight the shift from sapronotic to zoonotic transmission.

Page 3, Lines 24-25: This sentence should be more assertive, not only suggesting, but stating that “cats are the primary animal host for S. brasiliensis and vector of infection”. The same applies for line 138, that suggests dogs are not a major source of S. brasiliensis to humans. Even if cases of dog to human transmission went unnoticed for any reason, they definitely do not have a significant epidemiological role in this scenario.

R. Page 3, Line 130, has been edited to “Cats are the primary animal host…” to be more assertive in this statement. Page 4, Line 1454 has been edited to “Dogs are not a major source…” for a similar reason.

Page 4, Lines 163-164: The sensitivity for detecting Sporothrix in lesions of canine sporotrichosis is generally low, but it’s reported as high as 40%. Still, the fungal load is so low that makes the transmission unlikely.

R. On Page 4, Line 170-171, minor edits were made to clarify that the study in dogs provided evidence for a low likelihood of transmission.

Page 6, Line 247: Not only Tapeworm, I would suggest helminthic diseases in general, as anti-helminthic immunity may antagonize or suppress the immune response to other pathogens.

R. Page 6, Line 255 has been edited from “tapeworm coinfection” to “helminth coinfections” as suggested.

Page 8, Line 298: Please, provide more information on the adequate specimen for each kind of diagnostic method within the text or embedded in the table.

R. Table 1 (Page 9) was updated with an additional column for the “Specimen Type” required for each diagnostic technique.

Page 8, Lines 316-334: This paragraph needs more clarification and clearly state that yeast cells can be detected either from cytological preparations or biopsy specimens. Asteroid bodies are not common in lesions of human sporotrichosis from the Brazilian outbreaks and this issue is not really on the scope of the reference provided unless there was some referencing mistake.

R. Page 8, Line 326 has been edited to clarify that direct examination can be used for both cytological preparations and biopsy specimens. The reference for asteroid bodies was incorrect and has been updated with the appropriate reference at Line 342.

Page 10, Lines 356-359: Please, consider including antifungal susceptibility assays into the treatment component.

R. Thank you for your comment. In the antifungal susceptibility studies carried out in our co-author’s laboratory (Lapclin-Dermzoo, INI, Fiocruz), they observed no association between the result of the in vitro antifungal susceptibility test and the treatment outcome (unpublished results). Therefore, we have chosen not to include AFST in the treatment component.

Page 11, Line 383: This section is lacking some references, especially on the first and second paragraphs  

R. Additional references were added to the “Preventing cat to human transmission” section starting on Page 11, Line 397.

Page 12, Lines 482-483: I don’t think the ability to persist in the natural environment was an explored feature in this paper, as I don’t see this as a particularity of S. brasiliensisSporothrix brasiliensis, as pointed in this paper, seems much more adapted and hard to control in the feline host.

R. Page 12, Line 499-500, has been edited to replace “ability to persist in the natural environment” to “adaptation to the feline host.” This emphasizes that the difficulty in controlling this pathogen is due to the adaptation to the feline host, rather than persistence in the environment.

Reviewer 2 Report

The authors have done extensive literature review and tried to highlight the importance of cat-transmitted sporotrichosis (CTS) in south America as an emerging zoonoses at the animal-human interface. The etiology, mode of transmission and clinical manifestation of cat-transmitted sporotrichosis have been described systematically. The existing treatment regimens and availability and affordability of antifungal drugs and its application in low-income settings have been well documented. Simple and molecular diagnostic tools including antifungal sensitivity test have been described which is important in the context of emerging AMR problem. Challenges in diagnosis, prevention and control of CTS have been highlighted. The authors have voiced one health approach to be applied but it will be good to highlight what kinds of collaboration are needed between medical and veterinary professionals, i.e. joint investigation, joint surveillance, joint training or research etc.. It will be good if the authors will shed light on vaccine development? It will be necessary to give photo credit.

Author Response

Reviewer 2 Comments:

The authors have voiced one health approach to be applied but it will be good to highlight what kinds of collaboration are needed between medical and veterinary professionals, i.e. joint investigation, joint surveillance, joint training or research etc.

R. On Page 12, starting at Line 449, a sentence has been added to state, “Physicians and veterinarians should also collaborate on the investigation of suspected zoonotic outbreaks, development and enhancement of surveillance systems for human and feline sporotrichosis, and research into One Health techniques for controlling CTS.”

It will be good if the authors will shed light on vaccine development?

R. A sentence has been added to the “Prevention and Control” section on Page 11, Line 392-393 to state “An effective vaccine for human or feline sporotrichosis does not yet exist, but development of such a vaccine could be instrumental in the control of CTS.”

It will be necessary to give photo credit.

R. Photo credit has been added to Figures 1 and 2, on Pages 5 & 7.